# Adversarial Attacks on Graph Neural Networks via Meta Learning

**Daniel Zügner and Stephan Günnemann**
Technical University of Munich, Germany
{zuegnerd,guennemann}@in.tum.de

## Abstract

Deep learning models for graphs have advanced the state of the art on many tasks. Despite their recent success, little is known about their robustness. We investigate training time attacks on graph neural networks for node classification that perturb the discrete graph structure. Our core principle is to use meta-gradients to solve the bilevel problem underlying training-time attacks, essentially treating the graph as a hyperparameter to optimize. Our experiments show that small graph perturbations consistently lead to a strong decrease in performance for graph convolutional networks, and even transfer to unsupervised embeddings. Remarkably, the perturbations created by our algorithm can misguide the graph neural networks such that they perform *worse* than a simple baseline that ignores all relational information. Our attacks do not assume any knowledge about or access to the target classifiers.

## 1 Introduction

Graphs are a powerful representation that can model diverse data from virtually any domain, such as biology (protein interaction networks), chemistry (molecules), or social networks (Facebook). Not surprisingly, machine learning on graph data has a longstanding history, with tasks ranging from node classification, over community detection, to generative modeling.

In this paper, we study node classification, which is an instance of semi-supervised classification: given a single (attributed) network and a subset of nodes whose class labels are known (e.g., the topic of a paper in a citation graph), the goal is to infer the classes of the unlabeled nodes. While there exist many classical approaches to node classification (London & Getoor, 2014; Chapelle et al., 2006), recently *deep learning* on graphs has gained much attention (Monti et al., 2017; Bojchevski & Günnemann, 2018a; Battaglia et al., 2018; Perozzi et al., 2014; Bojchevski et al., 2018; Klicpera et al., 2019). Specifically, graph convolutional approaches (Kipf & Welling, 2017; Pham et al., 2017) have improved the state of the art in node classification.

However, recent works have also shown that such approaches are vulnerable to adversarial attacks both at test time (*evasion*) as well as training time (*poisoning* attacks) (Zügner et al., 2018; Dai et al., 2018). A core strength of models using graph convolution – exploiting the information in a node's neighborhood to improve classification – is also a major vulnerability: because of these propagation effects, an attacker can change a single node's prediction without even changing any of its attributes or edges. This is because the foundational assumption that all samples are *independent* of each other does not hold for node classification. Network effects such as homophily (London & Getoor, 2014) support the classification, while on the other hand they enable indirect adversarial attacks.

So far, all existing attacks on node classification models are *targeted*, that is, aim to provoke misclassification of a specific single node, e.g. a person in a social network. In this work, we propose the first algorithm for poisoning attacks that is able to compromise the *global* node classification performance of a model. We show that even under restrictive attack settings and without access to the target classifier, our attacks can render it near-useless for use in production (i.e., on test data).

Our approach is based on the principle of meta learning, which has traditionally been used for hyperparameter optimization (Bengio, 2000), or, more recently, few-shot learning (Finn et al., 2017). In essence, we turn the gradient-based optimization procedure of deep learning models upside down and treat the input data – the graph at hand – as a hyperparameter to learn.

## 2   RELATED WORK

Adversarial attacks on machine learning models have been studied both in the machine learning and security community and for many different model types (Mei & Zhu, 2015). It is important to distinguish attacks from outliers; while the latter naturally occur in graphs (Bojchevski & Günnemann, 2018), adversarial examples are deliberately created with the goal to mislead machine learning models and often designed to be unnoticeable. Deep neural networks are highly sensitive to these small adversarial perturbations to the data (Szegedy et al., 2014; Goodfellow et al., 2015). The vast majority of attacks and defenses assume the data instances to be independent and continuous. This assumption clearly does not hold for node classification and many other tasks on graphs.

Works on adversarial attacks for graph learning tasks are generally sparse. Chen et al. (2017) have measured the changes in the resulting graph clustering when injecting noise to a bi-partite graph that represent DNS queries. However, their focus is not on generating attacks in a principled way. Torkamani & Lowd (2013) consider adversarial noise in the node features in order to improve robustness of collective classification via associative Markov networks.

Only recently researchers have started to study adversarial attacks on deep learning for graphs. Dai et al. (2018) consider *test-time* (i.e., evasion) attacks on graph classification (i.e., classification of graphs themselves) and node classification. However, they do not consider poisoning (i.e., training-time) attacks or evaluate transferability of their attacks, and restrict the attacks to edge deletions only. Moreover, they focus on targeted attacks, i.e. attacks designed to change the prediction of a single node. Zügner et al. (2018) consider both test-time and training-time attacks on node classification models. They circumvent explicitly tackling the bilevel optimization problem underlying poisoning attacks by performing their attacks based on a (static) surrogate model and evaluating their impact by training a classifier on the data modified by their algorithm. In contrast to Dai et al. (2018), their attacks can both insert and remove edges, as well as modify node attributes in the form of binary vectors. Again, their algorithm is suited only to targeted attacks on single nodes; the problem of training-time attacks on the overall performance of node classification models remains unexplored. Bojchevski & Günnemann (2018b) propose poisoning attacks on a different task: unsupervised node representation learning (or node embeddings). They exploit perturbation theory to maximize the loss obtained after training DeepWalk. In this work, we focus on semi-supervised learning.

Meta-learning (Thrun & Pratt, 1998; Naik & Mammone, 1992), or *learning to learn*, is the task of optimizing the learning algorithm itself; e.g., by optimizing the hyperparameters Bengio (2000), learning to update the parameters of a neural network (Schmidhuber, 1992; Bengio et al., 1992), or the activation function of a model (Agostinelli et al., 2014). Gradient-based hyperparameter optimization works by differentiating the training phase of a model to obtain the gradients w.r.t. the hyperparameters to optimize.

The key idea of this work is to use meta-learning for the opposite: modifying the training data to *worsen* the performance after training (i.e., training-time or poisoning attacks). Muñoz-González et al. (2017) demonstrate that meta learning can indeed be used to create training-time attacks on simple, linear classification models. On continuous data, they report little success when attacking deep neural networks, and on discrete datasets, they do not consider deep learning models or problems with more than two classes. Like most works on adversarial attacks, they assume the data instances to be independent. In this work, for the first time, we propose an algorithm for global attacks on (deep) node classification models at training time. In contrast to Zügner et al. (2018), we explicitly tackle the bilevel optimization problem of poisoning attacks using meta learning.

## 3   PROBLEM FORMULATION

We consider the task of (semi-supervised) node classification. Given a single (attributed) graph and a set of labeled nodes, the goal is to infer the class labels of the unlabeled nodes. Formally, let $G = (A, X)$ be an attributed graph with adjacency matrix $A \in \{0, 1\}^{N \times N}$ and node attribute matrix $X \in \mathbb{R}^{N \times D}$, where $N$ is the number of nodes and $D$ the dimension of the node feature vectors. W.l.o.g., we assume the node IDs to be $\mathcal{V} = \{1, \ldots, N\}$.

Given the set of labeled nodes $\mathcal{V}_L \subseteq \mathcal{V}$, where nodes are assigned exactly one class in $\mathcal{C} = \{c_1, c_2, ..., c_K\}$, the goal is to learn a function $f_\theta$, which maps each node $v \in \mathcal{V}$ to exactly one

of the $K$ classes in $\mathcal{C}$ (or in a probabilistic formulation: to the K-simplex). Note that this is an instance of transductive learning, since *all* test samples (i.e., the unlabeled nodes) as well as their attributes and edges (but not their class labels!) are known and used during training (Chapelle et al., 2006). The parameters $\theta$ of the function $f_\theta$ are generally learned by minimizing a loss function $\mathcal{L}_{\text{train}}$ (e.g. cross-entropy) on the labeled training nodes:

$$\theta^* = \arg\min_\theta \mathcal{L}_{\text{train}}(f_\theta(G)), \tag{1}$$

where we overload the notation of $f_\theta$ to indicate that we feed in the whole graph $G$.

### 3.1 ATTACK MODEL

Adversarial attacks are small deliberate perturbations of data samples in order to achieve the outcome desired by the attacker when applied to the machine learning model at hand. The attacker is constrained in the knowledge they have about the data and the model they attack, as well as the adversarial perturbations they can perform.

**Attacker's goal.** In our work, the attacker's goal is to increase the misclassification rate (i.e., one minus the accuracy) of a node classification algorithm achieved after training on the data (i.e., graph) modified by our algorithm. In contrast to Zügner et al. (2018) and Dai et al. (2018), our algorithm is designed for *global* attacks reducing the overall classification performance of a model. That is, the goal is to have the test samples classified as *any* class different from the true class.

**Attacker's knowledge.** The attacker can have different levels of knowledge about the training data, i.e. the graph $G$, the target machine learning model $\mathcal{M}$, and the trained model parameters $\theta$. In our work, we focus on limited-knowledge attacks where the attacker has no knowledge about the classification model and its trained weights, but the same knowledge about the data as the classifier. In other words, the attacker can observe all nodes' attributes, the graph structure, as well as the labels of the subset $\mathcal{V}_L$ and uses a surrogate model to modify the data. Besides assuming knowledge about the full data, we also perform experiments where only a subset of the data is given. Afterwards, this modified data is used to train deep neural networks to degrade their performance.

**Attacker's capability.** In order to be effective and remain undiscovered, adversarial attacks should be *unnoticeable*. To account for this, we largely follow Zügner et al. (2018)'s attacker capabilities. First, we impose a budget constraint $\Delta$ on the attacks, i.e. limit the number of changes $\|A - \hat{A}\|_0 \le \Delta$ (here we have $2\Delta$ since we assume the graph to be symmetric). Furthermore, we make sure that no node becomes disconnected (i.e. a singleton) during the attack. One of the most fundamental properties of a graph is its degree distribution. Any significant changes to it are very likely to be noticed; to prevent such large changes to the degree distribution, we employ Zügner et al. (2018)'s unnoticeability constraint on the degree distribution. Essentially, it ensures that the graph's degree distribution can only marginally be modified by the attacker. The authors also derive an efficient way to check for violations of the constraint so that it adds only minimal computational overhead to the attacks. While in this work we focus on changing the graph structure only, our algorithm can easily be modified to change the node features as well. We summarize all these constraints and denote the set of admissible perturbations on the data as $\Phi(G)$, where $G$ is the graph at hand.

### 3.2 OVERALL GOAL

Poisoning attacks can be mathematically formulated as a bilevel optimization problem:

$$\min_{\hat{G} \in \Phi(G)} \mathcal{L}_{\text{atk}}(f_{\theta^*}(\hat{G})) \quad s.t. \quad \theta^* = \arg\min_\theta \mathcal{L}_{\text{train}}(f_\theta(\hat{G})). \tag{2}$$

$\mathcal{L}_{\text{atk}}$ is the loss function the attacker aims to optimize. In our case of global and unspecific (regarding the type of misclassification) attacks, the attacker tries to decrease the *generalization performance of the model on the unlabeled nodes*. Since the test data's labels are not available, we cannot directly optimize this loss. One way to approach this is to maximize the loss on the labeled (training) nodes $\mathcal{L}_{\text{train}}$, arguing that if a model has a high training error, it is very likely to also generalize poorly (the opposite is not true; when overfitting on the training data, a high generalization loss can correspond to a low training loss). Thus, our first attack option is to choose $\mathcal{L}_{\text{atk}} = -\mathcal{L}_{\text{train}}$.

Recall that semi-supervised node classification is an instance of *transductive* learning: all data samples (i.e., nodes) and their attributes are known at training time (but not all labels!). We can use

this insight to obtain a second variant of $\mathcal{L}_{\text{atk}}$. The attacker can learn a model on the labeled data to estimate the labels $\hat{C}_U$ of the unlabeled nodes $\mathcal{V}_U = \mathcal{V} \backslash \mathcal{V}_L$. The attacker can now perform self-learning, i.e. use these predicted labels and compute the loss of a model on the unlabeled nodes, yielding our second option $\mathcal{L}_{\text{atk}} = -\mathcal{L}_{\text{self}}$ where $\mathcal{L}_{\text{self}} = \mathcal{L}(\mathcal{V}_U, \hat{C}_U)$. Note that, at all times, only the labels of the labeled nodes are used for training; $\mathcal{L}_{\text{self}}$ is only used to estimate the generalization loss after training. In our experimental evaluation, we compare both versions of $\mathcal{L}_{\text{atk}}$ outlined above.

Importantly, notice the bilevel nature of the problem formulation in Eq. (2): the attacker aims to maximize the classification loss achieved *after* optimizing the model parameters on the modified (poisoned) graph $\hat{G}$. Optimizing such a bilevel problem is highly challenging by itself. Even worse, in our graph setting the data and the action space of the attacker are *discrete*: the graph structure is $A = \{0, 1\}^{N \times N}$, and the possible actions are edge insertions and deletions. This makes the problem even more difficult in two ways. First, the action space is vast; given a budget of $\Delta$ perturbations, the number of possible attacks is, ignoring symmetry, $\binom{N^2}{\Delta}$ and thus in $O(N^{2\Delta})$; exhaustive search is clearly infeasible. Second, a discrete data domain means that we cannot use gradient-based methods such as gradient descent to make *small* (real-valued) updates on the data to optimize a loss.

## 4 Graph Structure Poisoning via Meta-Learning

### 4.1 Poisoning via Meta-gradients

In this work, we tackle the bilevel problem described in Eq. (2) using meta-gradients, which have traditionally been used in meta-learning. The field of meta-learning (or learning to learn) tries to make the process of learning machine learning models more time and/or data efficient, e.g. by finding suitable hyperparameter configurations (Bengio, 2000) or initial weights that enable rapid adaptation to new tasks or domains in few-shot learning (Finn et al., 2017).

Meta-gradients (e.g., gradients w.r.t. hyperparameters) are obtained by backpropagating *through* the learning phase of a differentiable model (typically a neural network). The core idea behind our adversarial attack algorithm is **to treat the graph structure matrix as a hyperparameter** and compute the gradient of the attacker's loss *after training* with respect to it:

$$\nabla_G^{\text{meta}} := \nabla_G \mathcal{L}_{\text{atk}}(f_{\theta^*}(G)) \quad s.t. \quad \theta^* = \text{opt}_\theta(\mathcal{L}_{\text{train}}(f_\theta(G))), \tag{3}$$

where $\text{opt}(\cdot)$ is a differentiable optimization procedure (e.g. gradient descent and its stochastic variants) and $\mathcal{L}_{\text{train}}$ the training loss. Notice the similarity of the meta-gradient to the bi-level formulation in Eq. (2); the meta-gradient indicates how the attacker loss $\mathcal{L}_{\text{atk}}$ *after training* will change for small perturbations on the data, which is exactly what a poisoning attacker needs to know.

As an illustration, consider an example where we instantiate $\text{opt}$ with vanilla gradient descent with learning rate $\alpha$ starting from some initial parameters $\theta_0$

$$\theta_{t+1} = \theta_t - \alpha \nabla_{\theta_t} \mathcal{L}_{\text{train}}(f_{\theta_t}(G)) \tag{4}$$

The attacker's loss after training for $T$ steps is $\mathcal{L}_{\text{atk}}(f_{\theta_T}(G))$. The meta-gradient can be expressed by unrolling the training procedure:

$$\nabla_G^{\text{meta}} = \nabla_G \mathcal{L}_{\text{atk}}(f_{\theta_T}(G)) = \nabla_f \mathcal{L}_{\text{atk}}(f_{\theta_T}(G)) \cdot [\nabla_G f_{\theta_T}(G) + \nabla_{\theta_T} f_{\theta_T}(G) \cdot \nabla_G \theta_T], \text{ where} \tag{5}$$
$$\nabla_G \theta_{t+1} = \nabla_G \theta_t - \alpha \nabla_G \nabla_{\theta_t} \mathcal{L}_{\text{train}}(f_{\theta_t}(G))$$

Note that the parameters $\theta_t$ itself depend on the graph $G$ (see Eq. 4); they are *not fixed*. Thus, the derivative w.r.t. the graph has to be taken into account, chaining back until $\theta_0$. Given this, the attacker can use the meta-gradient to perform a meta update $M$ on the data to minimize $\mathcal{L}_{\text{atk}}$:

$$G^{(k+1)} \leftarrow M(G^{(k)}) \tag{6}$$

The final poisoned data $G^{(\Delta)}$ is obtained after performing $\Delta$ meta updates. A straightforward way to instantiate $M$ is (meta) gradient descent with some step size $\beta$: $M(G) = G - \beta \nabla_G \mathcal{L}_{\text{atk}}(f_{\theta_T}(G)))$.

It has to be noted that such a gradient-based update rule is neither possible nor well-suited for problems with discrete data (such as graphs). Due to the discreteness, the gradients are not defined. Thus, in our approach we simply relax the data's discreteness condition. However, we still perform discrete updates (actions) since the above simple gradient update would lead to dense (and continuous) adjacency matrices; not desired and not efficient to handle. Thus, in the following section, we propose a greedy approach to preserve the data's sparsity and discreteness.

## 4.2 GREEDY POISONING ATTACKS VIA META GRADIENTS

We assume that the attacker does not have access to the target classifier's parameters, outputs, or even knowledge about its architecture; the attacker thus uses a *surrogate* model to perform the poisoning attacks. Afterwards the poisoned data is used to train deep learning models for node classification (e.g. a GCN) to evaluate the performance degradation due to the attack. We use the same surrogate model as Zügner et al. (2018), which is a linearized two-layer graph convolutional network:

$$f_\theta(A, X) = \text{softmax}(\hat{A}^2 X W), \tag{7}$$

where $\hat{A} = D^{-1/2} \tilde{A} D^{-1/2}$, $\tilde{A} = A + I$, $A$ is the adjacency matrix, $X$ are the node features, $D$ the diagonal matrix of the node degrees, and $\theta = \{W\}$ the set of learnable parameters. In contrast to Zügner et al. (2018) we do not linearize the output (softmax) layer.

Note that we only perform changes to the graph structure $A$, hence we treat the node attributes $X$ as a constant during our attacks. For clarity, we replace $G$ with $A$ in the meta gradient formulation.

We define a score function $S : \mathcal{V} \times \mathcal{V} \to \mathbb{R}$ that assigns each possible action a numerical value indicating its (estimated) impact on the attacker objective $\mathcal{L}_{\text{atk}}$. Given the meta-gradient for a node pair $(u, v)$, we define $S(u, v) = \nabla_{a_{uv}}^{\text{meta}} \cdot (-2 \cdot a_{uv} + 1)$ where $a_{uv}$ is the entry at position $(u, v)$ in the adjacency matrix $A$. We essentially flip the sign of the meta-gradients for connected node pairs as this yields the gradient for a change in the negative direction (i.e., removing the edge).

We greedily pick the perturbation $e' = (u', v')$ with the highest score one at a time

$$e' = \underset{e=(u,v): M(A,e) \in \Phi(G)}{\arg\max} S(u, v), \tag{8}$$

where $M(A, e) \in \Phi(G)$ ensures that we only perform changes compliant with our attack constraints (e.g., unnoticeability). The meta update function $M(A, e)$ inserts the edge $e = (i, j)$ by setting $a_{ij} = 1$ if nodes $(i, j)$ are currently not connected and otherwise deletes the edge by setting $a_{ij} = 0$.

## 4.3 APPROXIMATING META-GRADIENTS

Computing the meta gradients is expensive both from a computational and a memory point-of-view. To alleviate this issue, Finn et al. (2017) propose a first-order approximation, leading to

$$\nabla_A^{\text{meta}} = \nabla_A \mathcal{L}_{\text{atk}}(f_{\theta_T}(A)) \approx \nabla_A \mathcal{L}_{\text{atk}}(f_{\tilde{\theta}_T}(A)) = \nabla_f \mathcal{L}_{\text{atk}}(f_{\tilde{\theta}_T}(A)) \cdot \nabla_A f_{\tilde{\theta}_T}(A). \tag{9}$$

We denote by $\tilde{\theta}_t$ the parameters at time $t$ *independent* of the data $A$ (and $\tilde{\theta}_{t-1}$), i.e. $\nabla_A \tilde{\theta}_t = 0$; the gradient is thus not propagated through $\tilde{\theta}_t$. This corresponds to taking the gradient of the attack loss $\mathcal{L}_{\text{atk}}$ w.r.t. the data, after training the model for $T$ steps. We compare against this baseline in our experiments; as also done in Zügner et al. (2018). However, unlike the meta-gradient, this approximation completely disregards the training dynamics.

Nichol & Schulman (2018) propose a heuristic of the meta gradient in which they update the initial weights $\theta_0$ on a line towards the local optimum $\theta_T$ to achieve faster convergence in a multi-task learning setting: $\nabla_{\theta_0}^{\text{meta}} \approx \sum_{t=1}^{T} \nabla_{\tilde{\theta}_t} \mathcal{L}_{\text{train}}(f_{\tilde{\theta}_t}(A; X))$. Again, they assume $\tilde{\theta}_t$ to be independent of $\tilde{\theta}_{t-1}$. While there is no direct connection to the formulation of the meta gradient in Eq. (5), there is an intuition behind it: the heuristic meta gradient is the direction, in which, *on average*, we have observed the strongest increase in the training loss during the training procedure. The authors' experimental evaluation further indicates that this heuristic achieves similar results as the meta gradient while being much more efficient to compute (see Appendix C for a discussion on complexity).

Adapted to our adversarial attack setting on graphs, we get $\nabla_A^{\text{meta}} \approx \sum_{t=1}^{T} \nabla_A \mathcal{L}_{\text{train}}(f_{\tilde{\theta}_t}(A; X))$. We can view this as a heuristic of the meta gradient when $\mathcal{L}_{\text{atk}} = -\mathcal{L}_{\text{train}}$. Likewise, again taking the transductive learning setting into account, we can use self-learning to estimate the loss on the unlabeled nodes, replacing $\mathcal{L}_{\text{train}}$ by $\mathcal{L}_{\text{self}}$. Indeed, we combine these two views

$$\nabla_A^{\text{meta}} \approx \Sigma_{t=1}^{T} \lambda \nabla_A \mathcal{L}_{\text{train}}(f_{\tilde{\theta}_t}(A; X)) + (1 - \lambda) \nabla_A \mathcal{L}_{\text{self}}(f_{\tilde{\theta}_t}(A; X)), \tag{10}$$

where $\lambda$ can be used to weight the two objectives. This approximation has a much smaller memory footprint than the exact meta gradient since we don't have to store the whole training trajectory $\tilde{\theta}_1, \dots, \tilde{\theta}_T$ in memory; additionally, there there are no second-order derivatives to be computed. A summary of our algorithm can be found in Appendix A.

Table 1: Misclassification rate (in %) for different meta-gradient heuristics with 5% perturbed edges.

| | CORA-ML | | CITESEER | |
| --- | --- | --- | --- | --- |
| | GCN | CLN | GCN | CLN |
| Clean | $16.6 \pm 0.3$ | $17.3 \pm 0.3$ | $28.5 \pm 1.0$ | $28.3 \pm 0.8$ |
| A-Meta-Train | $21.2 \pm 0.9$ | $20.3 \pm 0.3$ | $31.8 \pm 0.8$ | $29.8 \pm 0.5$ |
| A-Meta-Self | $21.8 \pm 0.7$ | $18.9 \pm 0.3$ | $28.6 \pm 0.4$ | $28.5 \pm 0.4$ |
| A-Meta-Both | $22.5 \pm 0.6$ | $19.2 \pm 0.3$ | $28.9 \pm 0.4$ | $28.8 \pm 0.4$ |

Table 2: Misclassification rate (in %) with 5% perturbed edges.

| Attack | CORA | | | CITESEER | | | POLBLOGS | | | Avg. rank |
| --- | --- | --- | --- | --- | --- | --- | --- | --- | --- | --- |
| | GCN | CLN | DeepWalk | GCN | CLN | DeepWalk | GCN | CLN | DeepWalk | |
| Clean | $16.6 \pm 0.3$ | $17.3 \pm 0.3$ | $20.3 \pm 1.0$ | $28.5 \pm 0.9$ | $28.3 \pm 0.9$ | $34.8 \pm 1.4$ | $6.4 \pm 0.6$ | $7.6 \pm 0.5$ | $5.3 \pm 0.5$ | 7.4 |
| DICE | $18.0 \pm 0.4$ | $18.0 \pm 0.2$ | $22.8 \pm 0.3$ | $28.9 \pm 0.3$ | $29.1 \pm 0.3$ | $39.1 \pm 0.4$ | $11.2 \pm 1.1$ | $11.2 \pm 0.8$ | $10.2 \pm 0.6$ | 5.0 |
| First-order | $17.2 \pm 0.3$ | $17.6 \pm 0.2$ | $20.7 \pm 0.2$ | $28.3 \pm 0.3$ | $28.4 \pm 0.3$ | $34.0 \pm 0.3$ | $7.8 \pm 0.9$ | $7.6 \pm 0.5$ | $7.9 \pm 0.6$ | 7.1 |
| Nettack* | - | - | - | $31.9 \pm 0.3$ | $30.2 \pm 0.4$ | $\mathbf{41.2 \pm 0.4}$ | - | - | - | - |
| A-Meta-Train | $21.8 \pm 0.9$ | $20.5 \pm 0.3$ | $25.0 \pm 0.6$ | $31.9 \pm 0.7$ | $30.1 \pm 0.5$ | $32.7 \pm 0.5$ | $11.9 \pm 2.8$ | $12.9 \pm 2.5$ | $5.8 \pm 0.2$ | 4.7 |
| A-Meta-Both | $20.7 \pm 0.4$ | $19.0 \pm 0.3$ | $\mathbf{28.5 \pm 0.5}$ | $28.6 \pm 0.4$ | $28.7 \pm 0.4$ | $34.4 \pm 0.4$ | $19.8 \pm 0.8$ | $16.5 \pm 1.3$ | $21.5 \pm 1.9$ | 4.3 |
| Meta-Train | $22.0 \pm 1.2$ | $\mathbf{21.7 \pm 0.4}$ | $26.1 \pm 0.6$ | $30.3 \pm 1.0$ | $29.0 \pm 0.6$ | $36.0 \pm 0.2$ | $16.3 \pm 2.9$ | $\mathbf{18.7 \pm 2.3}$ | $14.5 \pm 4.2$ | 3.2 |
| Meta-Self | $\mathbf{24.5 \pm 1.0}$ | $20.3 \pm 0.4$ | $28.1 \pm 0.6$ | $\mathbf{34.6 \pm 0.7}$ | $\mathbf{32.2 \pm 0.6}$ | $34.6 \pm 0.7$ | $\mathbf{22.5 \pm 0.8}$ | $17.9 \pm 1.7$ | $\mathbf{59.0 \pm 3.0}$ | 2.3 |
| Meta w/ Oracle | $21.0 \pm 0.5$ | $21.6 \pm 0.3$ | $27.8 \pm 0.7$ | $34.2 \pm 0.9$ | $32.9 \pm 0.6$ | $36.1 \pm 0.7$ | $25.6 \pm 1.9$ | $19.1 \pm 1.4$ | $52.3 \pm 2.8$ | 2.0 |

* Did not finish within three days on CORA-ML and POLBLOGS

## 5 EXPERIMENTS

**Setup.** We evaluate our approach on the well-known CITESEER (Sen et al., 2008), CORA-ML (McCallum et al., 2000), and POLBLOGS (Adamic & Glance, 2005) datasets; an overview is given in Table 6. We split the datasets into labeled (10%) and unlabeled (90%) nodes. The labels of the unlabeled nodes are never visible to the attacker or the classifiers and are only used to evaluate the generalization performance of the models. Our code is available at `https://www.kdd.in.tum.de/gnn-meta-attack`.

We evaluate the transferability of adversarial attacks by training deep node classification models on the modified (poisoned) data. For this purpose, we use Graph Convolutional Networks (GCN) (Kipf & Welling, 2017) and Column Networks (CLN) (Pham et al., 2017). Both are models utilizing the message passing framework (a.k.a. graph convolution) and trained in a semi-supervised way. We further evaluate the node classification performance achieved by training a standard logistic regression model on the node embeddings learned by DeepWalk (Perozzi et al., 2014). DeepWalk itself is trained in an unsupervised way and without node attributes or graph convolutions; thus, this is arguably an even more difficult transfer task.

We repeat all of our attacks on five different splits of labeled/unlabeled nodes and train all target classifiers ten times per attack (using the split that was used to create the attack). In our tables, the uncertainty indicates $95\%$ confidence intervals of the mean obtained via bootstrapping. For our meta-gradient approaches, we compute the meta-gradient $\nabla_A^{\mathrm{meta}} \mathcal{L}_{\mathrm{atk}}(f_{\theta_T}(A; X))$ by using gradient descent with momentum for 100 iterations. We refer to our meta-gradient approach with self-training as Meta-Self and to the variant without self-training as Meta-Train. Similarly, we refer to our approximations as A-Meta-Self (with $\lambda=0$), A-Meta-Train ($\lambda=1$), and A-Meta-Both ($\lambda=0.5$).

**Comparing meta-gradient heuristics.** First, we analyze the different meta gradient heuristics described in Section 4.3. The results can be seen in Table 1. All principles successfully increase the misclassification rate (i.e., $1 -$ accuracy on unlabeled nodes) obtained on the test data, compared to the results obtained with the unperturbed graph. Since A-Meta-Self consistently shows a weaker performance than A-Meta-Both, we do not further consider A-Meta-Self in the following.

**Comparison with competing methods.** We compare our meta-gradient approach as well as its approximations with various baselines and Nettack (Zügner et al., 2018). DICE ('delete internally, connect externally') is a baseline where, for each perturbation, we randomly choose whether to insert or remove an edge. Edges are only removed between nodes from the same class, and only inserted between nodes from different classes. This baseline has *all* true class labels (train and test) available and thus more knowledge than all competing methods. First-order refers to the approximation proposed by Finn et al. (2017), i.e. ignoring all second-order derivatives. Note that Nettack is *not* designed for global attacks. In order to be able to compare to them, for each perturbation we ran-

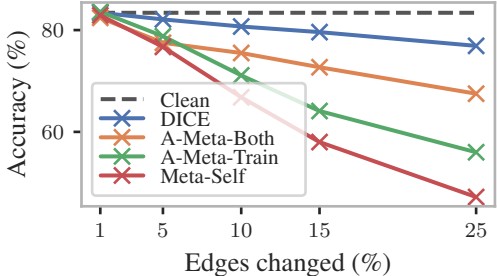 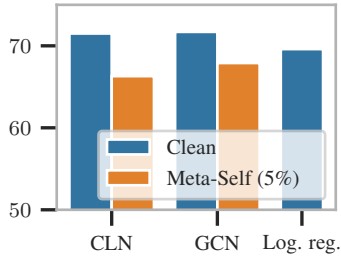

Figure 1: Change in accuracy of GCN on CORA-ML for increasing number of perturbations.

Figure 2: Comparison with logistic regression baseline on CITESEER.

Table 3: Accuracy of clean/ corrupted graph and weights.

Table 4: Poisoning results with limited knowledge about the graph (i.e. on a subgraph) after 10% changes.

|  | $W$ | $\hat{W}$ |
|---|---|---|
| $A$ | 0.85 | 0.52 |
| $\hat{A}$ | 0.83 | 0.49 |

|  | CORA-ML | | CITESEER | |
|---|---|---|---|---|
|  | GCN | CLN | GCN | CLN |
| Clean | $16.6 \pm 0.3$ | $17.3 \pm 0.3$ | $28.5 \pm 0.8$ | $28.3 \pm 0.8$ |
| A-Meta-Sub | $21.4 \pm 0.7$ | $22.4 \pm 0.4$ | $30.9 \pm 0.7$ | $31.4 \pm 0.7$ |
| Meta-Sub | $21.2 \pm 0.6$ | $20.8 \pm 0.3$ | $28.7 \pm 0.3$ | $31.4 \pm 0.5$ |

domly select one target node from the unlabeled nodes and attack it using Nettack while considering *all* nodes in the network. In this case, its time and memory complexity is $O(N^3)$ and thus it was not feasible to run it on any but the sparsest dataset. Meta w/ Oracle corresponds to our meta-gradient approach when supplied with *all* true class labels on the test data – this only serves as a reference point since it cannot be carried out in real scenarios where the test nodes' labels are unknown. For all methods, we enforce the *unnoticeability* constraint introduced by Zügner et al. (2018), which ensures that the graph's degree distribution changes only slightly. In Appendix D we show that the unnoticeability constraint does not significantly limit the impact of our attacks.

In Table 2 we see the misclassification rates (i.e., 1 - accuracy on unlabeled nodes) achieved by changing 5% of $E_{LCC}$ edges according to the different methods (larger is better, except for the average rank). That is, each method is allowed to modify 5% of $E_{LCC}$, i.e. the number of edges present in the graph before the attack. We present similar tables for 1% and 10% changes in Appendix F. Our meta-gradient with self-training (Meta-Self) produces the strongest drop in performance across all models and datasets as indicated by the *average rank*. Changing only 5% of the edges leads to a relative increase of up to 48% in the misclassification rate of GCN on CORA-ML.

Remarkably, our memory efficient meta-gradient approximations lead to strong increases in misclassifications as well. They outperform both baselines and are in many cases even on par with the more expensive meta-gradient. In Appendix F, Table 11 we also show that using only $T = 10$ training iterations of the surrogate models for computing the meta gradient (or its approximations) can significantly hurt the performance across models and datasets. Moreover, in Table 8 in Appendix F we show that our heuristic is successful at attacking a dataset with roughly 20K nodes.

While the focus of our work is poisoning attacks by modifying the graph structure, our method can be applied to node feature attacks as well. In Appendix E we show a proof of concept that our attacks are also effective when attacking by perturbing both node features and the graph structure.

In Fig. 1 we see the drop in classification performance of GCN on CORA-ML for increasing numbers of edge insertions/deletions (similar plots for the remaining datasets and models are provided in Appendix F). Meta-Self is even able to reduce the classification accuracy below 50%. Fig. 2 shows the classification accuracy of GCN and CLN as well as a baseline operating on the node attributes only, i.e. ignoring the graph. Not surprisingly, deep models achieve higher accuracy than the baseline when trained on the clean CITESEER graph – exploiting the network information improves classification. However, by only perturbing 5% of the edges, we obtain the opposite: GCN and CLN perform *worse* than the baseline – the graph structure now hurts classification.

**Impact of graph structure and trained weights.** Another interesting property of our attacks can be seen in Table 3, where $W$ and $\hat{W}$ correspond to the weights trained on the clean CORA-ML network $A$ and a version $\hat{A}$ poisoned by our algorithm (here with even 25 % modified edges), respectively.

Note that the classification accuracy barely changes when modifying the underlying network for a given set of trained weights; even when applying the clean weights $W$ on the highly corrupted $\hat{A}$, the performance drops only marginally. Likewise, even the *clean* graph $A$ only leads to a low accuracy when using it with the weights $\hat{W}$. This result emphasizes the importance of the training procedure for the performance of graph models and shows that our poisoning attack works by *derailing* the training procedure from the start, i.e. leading to 'bad' weights.

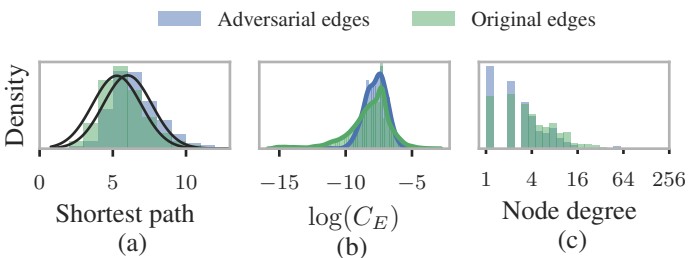

Table 5: Share (in %) of edge deletions (DEL) and insertions (INS) by Meta-Self on CORA-ML.

| | $c_i=c_j$ | $c_i\neq c_j$ |
|---|---|---|
| DEL | 15.3 | 3.9 |
| INS | 9.4 | 71.4 |

Figure 3: Analysis of adversarially inserted edges

**Analysis of attacks.** An interesting question to ask is *why* the adversarial changes created by our meta-gradient approach are so destructive, and what patterns they follow. If we can find out what makes an edge insertion or deletion a strong adversarial change, we can circumvent expensive meta-gradient computations or even use this knowledge to detect adversarial attacks.

In Fig. 3 we compare edges inserted by our meta-gradient approach to the edges originally present in the CORA-ML network. Fig. 3 (a) shows the shortest path lengths between nodes pairs *before* being connected by adversarially inserted edges vs. shortest path lengths between all nodes in the original graph. In Fig. 3 (b) we compare the edge betweenness centrality ($C_E$) of adversarially inserted edges to the centrality of edges present in the original graph. In (c) we see the node degree distributions of the original graph and the node degrees of the nodes that are picked for adversarial edges. For all three measures no clear distinction can be made. There is a slight tendency for the algorithm to connect nodes that have higher-than-average shortest paths and low degrees, though.

As we can see in Table 5, roughly 80% of our meta attack's perturbations are edge insertions (INS). As expected by the homophily assumption, in most cases edges inserted connect nodes from different classes and edges deleted connect same-class nodes. However, as the comparison with the DICE baseline shows, this by itself can also not explain the destructive performance of the meta-gradient.

**Limited knowledge about the graph structure.** In the experiments described above, the attacker has full knowledge about the graph structure and all node attributes (as typical in a transductive setting). We also tested our algorithm on a *sub-graph* of CORA-ML and CITESEER. That is, we select the 10% labeled nodes and randomly select neighbors of these until we have a subgraph with number of nodes $n = 0.3N$. We run our attacks on this small subgraph, and afterwards plug in the perturbations into the original graphs to train GCN and CLN as before. Table 4 summarizes the results: Even in this highly restricted setting, our attacks consistently increase misclassification rate across datasets and models, highlighting the effectiveness of our method.

## 6 CONCLUSION

We propose an algorithm for training-time adversarial attacks on (attributed) graphs, focusing on the task of node classification. We use meta-gradients to solve the bilevel optimization problem underlying the challenging class of poisoning adversarial attacks. Our experiments show that attacks created using our meta-gradient approach consistently lead to a strong decrease in classification performance of graph convolutional models and even transfer to unsupervised models. Remarkably, even small perturbations to a graph based on our approach can lead to graph neural networks performing worse than a baseline ignoring all relational information. We further propose approximations of the meta-gradients that are less expensive to compute and, in many cases, have a similarly destructive impact on the training of node classification models. While we are able to show small statistical differences of adversarial and 'normal' edges, it is still an open question what makes the edges inserted/removed by our algorithm so destructive, which could then be used to detect or defend against attacks.

ACKNOWLEDGEMENTS

This research was supported by the German Research Foundation, grant GU 1409/2-1.

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

## A  ALGORITHM

---

**Algorithm 1:** Poisoning attack on graph neural networks with meta gradients and self-training

---

**Input:** Graph $G = (A, X)$, modification budget $\Delta$, number of training iterations $T$, training class labels $C_L$

**Output:** Modified graph $\hat{G} = (\hat{A}, X)$

$\hat{\theta} \leftarrow$ train surrogate model on the input graph using known labels $C_L$;

$\hat{C}_U \leftarrow$ predict labels of unlabeled nodes using $\hat{\theta}$;

$\hat{A} \leftarrow A$;

**while** $\|\hat{A} - A\|_0 < 2\Delta$ **do**

    randomly initialize $\theta_0$;

    **for** $t$ *in* $0 \dots T - 1$ **do**

        $\theta_{t+1} \leftarrow$ step $(\theta_t, \nabla_{\theta_t} \mathcal{L}_{\text{train}}(f_{\theta_t}(\hat{A}, X)); C_L)$;       // update e.g. via gradient descent

    // Compute meta gradient via backprop through the training procedure

    $\nabla_{\hat{A}}^{\text{meta}} \leftarrow \nabla_{\hat{A}} \mathcal{L}_{\text{self}}(f_{\theta_T}(\hat{A}, X); \hat{C}_U)$;

    $S \leftarrow \nabla_{\hat{A}}^{\text{meta}} \odot (-2\hat{A} + 1)$;       // Flip gradient sign of node pairs with edge

    $e' \leftarrow$ maximum entry $(u, v)$ in $S$ that fulfills constraints $\Phi(G)$;

    $\hat{A} \leftarrow$ insert or remove edge $e'$ to/from $\hat{A}$;

$\hat{G} \leftarrow (\hat{A}, X)$;

**return** : $\hat{G}$

---

**Algorithm 2:** Poisoning attack on GNNs with approximate meta gradients and self-training

---

**Input:** Graph $G = (A, X)$, modification budget $\Delta$, number of training iterations $T$, gradient weighting $\lambda$, training class labels $C_L$

**Output:** Modified graph $\hat{G} = (\hat{A}, X)$

$\hat{\theta} \leftarrow$ train surrogate model on the input graph using known labels $C_L$;

$\hat{C}_U \leftarrow$ predict labels of unlabeled nodes using $\hat{\theta}$;

$\hat{A} \leftarrow A$;

**while** $\|\hat{A} - A\|_0 < 2\Delta$ **do**

    randomly initialize $\theta_0$;

    $\nabla_{\hat{A}}^{\text{meta}} \leftarrow \lambda \nabla_{\hat{A}} \mathcal{L}_{\text{train}}(f_{\theta_0}(\hat{A}; X); C_L) + (1 - \lambda) \nabla_{\hat{A}} \mathcal{L}_{\text{self}}(f_{\theta_0}(\hat{A}; X); \hat{C}_U)$

    **for** $t$ *in* $0 \dots T - 1$ **do**

        $\theta_{t+1} \leftarrow$ step $(\theta_t, \nabla_{\theta_t} \mathcal{L}_{\text{train}}(f_{\theta_t}(\hat{A}, X)); C_L)$;       // update e.g. via gradient descent

        $\tilde{\theta}_{t+1} \leftarrow$ stop_gradient$(\theta_{t+1})$;       // no backprop through training

        $\nabla_{\hat{A}}^{\text{meta}} \leftarrow \nabla_{\hat{A}}^{\text{meta}} + \lambda \nabla_{\hat{A}} \mathcal{L}_{\text{train}}(f_{\tilde{\theta}_{t+1}}(\hat{A}; X); C_L) + (1 - \lambda) \nabla_{\hat{A}} \mathcal{L}_{\text{self}}(f_{\tilde{\theta}_{t+1}}(\hat{A}; X); \hat{C}_U)$

    $S \leftarrow \nabla_{\hat{A}}^{\text{meta}} \odot (-2\hat{A} + 1)$;       // Flip gradient sign of node pairs with edge

    $e' \leftarrow$ maximum entry $(u, v)$ in $S$ that fulfills constraints $\Phi(G)$;

    $\hat{A} \leftarrow$ insert or remove edge $e'$ to/from $\hat{A}$;

$\hat{G} \leftarrow (\hat{A}, X)$;

**return** : $\hat{G}$

---

## B  DATASET STATISTICS

Table 6: Dataset statistics.

| Dataset | $N_{LCC}$ | $E_{LCC}$ | D | K |
|---------|-----------|-----------|-----|---|
| CORA-ML | 2,810 | 7,981 | 2,879 | 7 |
| CITESEER | 2,110 | 3,757 | 3,703 | 6 |
| POLBLOGS | 1,222 | 16,714 | - | 2 |
| PUBMED | 19,717 | 44,324 | 500 | 3 |

In Table 6 we see the characteristics of the datasets used in this work. Results for PUBMED can be found in Table 8 in Appendix F.

## C  COMPLEXITY ANALYSIS

In our attack we handle both edge insertions and deletions, i.e. each element in the adjacency matrix $A \in \{0, 1\}^{N \times N}$ can be changed. This means that without further optimization, the (approximate) meta gradient for each node pair has to be computed, leading to a baseline memory and computational complexity of $O(N^2)$. For the meta gradient computation we additionally have to store the entire weight trajectory during training, adding $O(T \cdot |\theta|)$ to the memory cost, where $T$ is the number of inner training steps and $|\theta|$ the number of weights. Thus, memory complexity of our meta gradient attack is $O(N^2 + T \cdot |\theta|)$. The second-order derivatives at each step $T$ in the meta gradient formulation can be computed in $O(N^2)$ using Hessian-vector products, leading to a computational complexity of $O(T \cdot N^2)$.

For the meta gradient heuristics, the computational complexity is similar since we have to evaluate the gradient w.r.t. the adjacency matrix at every training step. However, the training trajectory of the weights does not have to be kept in memory, yielding a memory complexity of $O(N^2)$. This is highly beneficial, as memory (especially on GPUs) is limited.

The computational and memory complexity of our adversarial attacks implies that (as-is) it can be executed for graphs with roughly $20K$ nodes using a commodity GPU. The complexity, however, can be drastically reduced by pre-filtering the elements in the adjacency matrix for which the (meta) gradient needs to be computed, since only a fraction of entries in the adjacency matrix are promising candidate perturbations. We leave such performance optimization for future work.

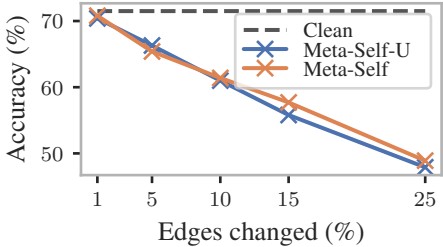

Figure 4: Change in accuracy of GCN on CITESEER with and without enforcing unnoticeability constraints (singleton nodes are never admissible). Meta-Self-U corresponds to not enforcing the unnoticeability constraint.

Table 7: Misclassification rate (in %) with 10% perturbations in edges / node features. For Meta-Self with features, at each step the perturbation (edge or feature) is selected that has the highest meta gradient score.

|  | CITESEER | |
|  | GCN | CLN |
|---|---|---|
| Clean | $28.5 \pm 0.9$ | $28.3 \pm 0.8$ |
| Meta-Self with features | $37.2 \pm 1.1$ | $34.2 \pm 0.7$ |
| Meta-Self | $38.6 \pm 1.0$ | $35.3 \pm 0.7$ |

## D  UNNOTICEABILITY CONSTRAINT

In all our experiments, we enforce the unnoticeability constraint on the degree distribution proposed by (Zügner et al., 2018). In Fig. 4 we show that this constraint does not significantly limit the destructive performance of our attacks. Thus we conclude that these constraints should always be enforced, since they improve unnoticeability while at the same time our attacks remain effective.

## E  ATTACKS WITH CHANGES TO THE NODE FEATURES

While the focus of our work is poisoning attacks by modifying the graph structure, our method can be applied to node feature attacks as well. The most straightforward case is when the node features are binary, since then we can use the same greedy algorithm as for the graph structure (ignoring the degree distribution constraints). Among the datasets we evaluated, CITESEER has binary node features, hence in Table 7 we display the results when attacking both node features and graph structure (while the total number of perturbations stays the same). We can observe that the impact of the combined attacks is slightly lower than the structure-only attack. We attribute this to the fact that we assign the same cost to structure and feature changes, but arguably we expect a structure perturbation to have a stronger effect on performance than a feature perturbation. Future work can provide a framework where structure and feature changes impose a different cost on the attacker. When the node features are continuous, there also needs to be some tuning of the meta step size and considerations whether multiple features per instance can be changed in a single step.

## F  ADDITIONAL RESULTS

In this section we present additional results of our experiments. In Table 8 we see that our heuristic is successful at attacking PUBMED, a dataset with roughly 20K nodes. Tables 9 and 10 show misclassification rates with 1% and 10% perturbed edges, respectively. Table 11 displays results when training the surrogate model for $T = 10$ iterations to obtain the (meta) gradients. Finally, Figures 5 through 12 show how the respective models' classification accuracies change for different attack methods and datasets.

Table 8: Misclassification rate (in %) with 5% perturbed edges on PUBMED Sen et al. (2008) when training the surrogate model for $T = 30$ iterations to compute the approximate meta gradients.

|  | PUBMED | | |
| --- | --- | --- | --- |
|  | GCN | CLN | DeepWalk |
| Clean | $13.8 \pm 0.3$ | $15.9 \pm 0.5$ | $21.8 \pm 0.1$ |
| DICE | $15.3 \pm 0.1$ | $16.6 \pm 0.4$ | $25.1 \pm 0.1$ |
| A-Meta-Self | $16.4 \pm 0.2$ | $16.4 \pm 0.4$ | $27.4 \pm 0.2$ |

Table 9: Misclassification rate (in %) with 1% perturbed edges.

| Attack | CORA | | | CITESEER | | | POLBLOGS | | | Avg. rank |
| --- | --- | --- | --- | --- | --- | --- | --- | --- | --- | --- |
|  | GCN | CLN | DeepWalk | GCN | CLN | DeepWalk | GCN | CLN | DeepWalk | |
| Clean | $16.6 \pm 0.3$ | $17.3 \pm 0.3$ | $20.3 \pm 0.9$ | $28.5 \pm 0.8$ | $28.3 \pm 0.8$ | $34.8 \pm 1.3$ | $6.4 \pm 0.5$ | $7.6 \pm 0.5$ | $5.3 \pm 0.5$ | 6.3 |
| DICE | $16.6 \pm 0.3$ | $17.6 \pm 0.2$ | $20.1 \pm 0.2$ | $28.4 \pm 0.3$ | $28.4 \pm 0.3$ | $35.9 \pm 0.3$ | $7.7 \pm 0.9$ | $8.5 \pm 0.6$ | $7.4 \pm 1.0$ | 4.8 |
| First-order | $16.6 \pm 0.3$ | $17.3 \pm 0.1$ | $20.2 \pm 0.2$ | $28.2 \pm 0.3$ | $28.3 \pm 0.4$ | $34.9 \pm 0.4$ | $7.0 \pm 0.8$ | $7.7 \pm 0.5$ | $8.5 \pm 1.6$ | 5.7 |
| Nettack* | - | - | - | $29.0 \pm 0.4$ | $28.6 \pm 0.4$ | $36.4 \pm 0.4$ | - | - | - | - |
| A-Meta-Train | $16.3 \pm 0.4$ | $18.2 \pm 0.2$ | $20.6 \pm 0.3$ | $29.1 \pm 0.5$ | $28.6 \pm 0.5$ | $34.7 \pm 0.5$ | $8.9 \pm 2.9$ | $10.2 \pm 1.9$ | $5.0 \pm 0.2$ | 4.7 |
| A-Meta-Both | $17.4 \pm 0.4$ | $17.6 \pm 0.2$ | $21.6 \pm 0.3$ | $28.5 \pm 0.4$ | $28.3 \pm 0.5$ | $34.6 \pm 0.3$ | $13.7 \pm 1.6$ | $10.4 \pm 1.2$ | $7.5 \pm 0.6$ | 3.6 |
| Meta-Train | $16.2 \pm 0.3$ | $18.0 \pm 0.3$ | $20.6 \pm 0.4$ | $28.3 \pm 0.5$ | $28.1 \pm 0.6$ | $35.3 \pm 0.3$ | $9.4 \pm 1.1$ | $10.3 \pm 1.7$ | $7.3 \pm 2.5$ | 4.8 |
| Meta-Self | $17.0 \pm 0.4$ | $17.9 \pm 0.2$ | $21.1 \pm 0.3$ | $29.2 \pm 0.5$ | $29.0 \pm 0.4$ | $35.2 \pm 0.3$ | $11.4 \pm 0.4$ | $10.7 \pm 1.7$ | $6.6 \pm 1.4$ | 2.7 |
| Meta with Oracle | $16.2 \pm 0.3$ | $18.2 \pm 0.2$ | $20.5 \pm 0.3$ | $30.1 \pm 0.5$ | $29.5 \pm 0.5$ | $34.8 \pm 0.3$ | $13.6 \pm 1.1$ | $10.5 \pm 1.2$ | $6.0 \pm 0.7$ | 3.5 |

* Did not finish within three days on CORA-ML and POLBLOGS

Table 10: Misclassification rate (in %) with 10% perturbed edges.

| Attack | CORA | | | CITESEER | | | POLBLOGS | | | Avg. rank |
| --- | --- | --- | --- | --- | --- | --- | --- | --- | --- | --- |
|  | GCN | CLN | DeepWalk | GCN | CLN | DeepWalk | GCN | CLN | DeepWalk | |
| Clean | $16.6 \pm 0.3$ | $17.3 \pm 0.3$ | $20.3 \pm 1.0$ | $28.5 \pm 0.8$ | $28.3 \pm 0.8$ | $34.8 \pm 1.3$ | $6.4 \pm 0.5$ | $7.6 \pm 0.5$ | $5.3 \pm 0.5$ | 7.5 |
| DICE | $19.5 \pm 0.5$ | $19.1 \pm 0.2$ | $26.2 \pm 0.3$ | $29.7 \pm 0.3$ | $29.9 \pm 0.3$ | $41.2 \pm 0.3$ | $14.4 \pm 0.8$ | $14.3 \pm 0.6$ | $12.9 \pm 0.3$ | 4.9 |
| First-order | $17.6 \pm 0.5$ | $17.9 \pm 0.2$ | $21.5 \pm 0.2$ | $28.2 \pm 0.3$ | $28.7 \pm 0.4$ | $32.4 \pm 0.4$ | $7.7 \pm 0.6$ | $7.6 \pm 0.3$ | $8.2 \pm 0.6$ | 7.1 |
| Nettack* | - | - | - | - | - | - | - | - | - | - |
| A-Meta-Train | $28.1 \pm 1.1$ | $23.6 \pm 0.4$ | $33.6 \pm 0.7$ | $34.3 \pm 1.1$ | $31.3 \pm 0.6$ | $32.1 \pm 0.5$ | $12.8 \pm 1.6$ | $18.2 \pm 2.6$ | $6.9 \pm 0.2$ | 4.9 |
| A-Meta-Both | $24.6 \pm 1.0$ | $20.0 \pm 0.3$ | $34.8 \pm 0.6$ | $29.1 \pm 0.5$ | $29.2 \pm 0.4$ | $33.6 \pm 0.4$ | $22.7 \pm 0.7$ | $22.3 \pm 0.9$ | $26.3 \pm 1.0$ | 4.8 |
| Meta-Train | $37.3 \pm 1.4$ | $24.9 \pm 0.5$ | $34.4 \pm 1.6$ | $31.8 \pm 1.0$ | $29.9 \pm 0.7$ | $36.0 \pm 0.2$ | $28.7 \pm 3.6$ | $32.9 \pm 1.6$ | $73.7 \pm 3.9$ | 2.6 |
| Meta-Self | $34.5 \pm 0.9$ | $22.9 \pm 0.6$ | $37.0 \pm 1.0$ | $38.6 \pm 1.0$ | $35.3 \pm 0.7$ | $36.0 \pm 1.2$ | $26.1 \pm 0.6$ | $23.5 \pm 0.9$ | $60.7 \pm 2.7$ | 2.8 |
| Meta with Oracle | $34.8 \pm 1.5$ | $25.2 \pm 0.4$ | $44.0 \pm 0.9$ | $40.1 \pm 1.2$ | $37.2 \pm 0.9$ | $37.2 \pm 0.6$ | $28.9 \pm 0.4$ | $25.8 \pm 0.9$ | $67.1 \pm 2.4$ | 1.4 |

* Did not finish within three days for any dataset.

Table 11: Misclassification rate (in %) with 5% perturbed edges and only training the surrogate model for $T = 10$ iterations to obtain the (meta-) gradients.

| Attack | CORA | | | CITESEER | | | POLBLOGS | | | Avg. rank |
| --- | --- | --- | --- | --- | --- | --- | --- | --- | --- | --- |
|  | GCN | CLN | DeepWalk | GCN | CLN | DeepWalk | GCN | CLN | DeepWalk | |
| Clean | $16.6 \pm 0.3$ | $17.3 \pm 0.3$ | $20.3 \pm 0.9$ | $28.5 \pm 0.8$ | $28.3 \pm 0.8$ | $34.8 \pm 1.3$ | $6.4 \pm 0.5$ | $7.6 \pm 0.5$ | $5.3 \pm 0.5$ | 3.0 |
| A-Meta-Both | $21.6 \pm 0.6$ | $18.9 \pm 0.3$ | $27.8 \pm 0.2$ | $31.6 \pm 0.4$ | $30.3 \pm 0.6$ | $40.7 \pm 0.4$ | $17.8 \pm 1.9$ | $13.9 \pm 1.4$ | $11.0 \pm 0.5$ | 1.8 |
| Meta-Self | $29.7 \pm 2.2$ | $20.1 \pm 0.4$ | $31.5 \pm 1.2$ | $29.9 \pm 0.7$ | $32.7 \pm 0.8$ | $45.6 \pm 0.7$ | $17.4 \pm 0.8$ | $14.6 \pm 1.2$ | $16.8 \pm 1.9$ | 1.2 |

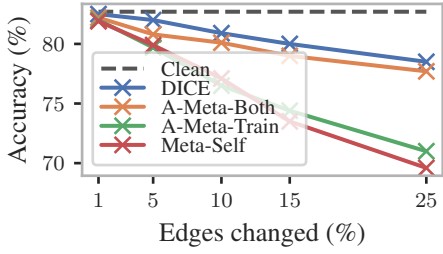

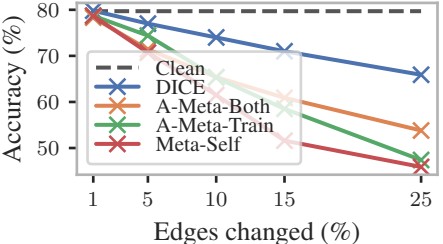

Figure 5: Change in accuracy of CLN on CORA-ML.

Figure 6: Change in accuracy of Deepwalk on CORA-ML.

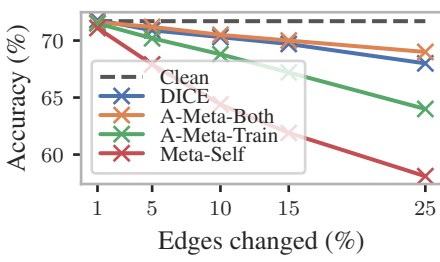

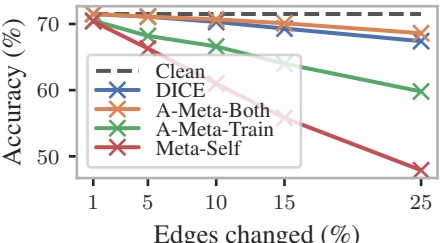

Figure 7: Change in accuracy of CLN on CITESEER.

Figure 8: Change in accuracy of GCN on CITESEER.

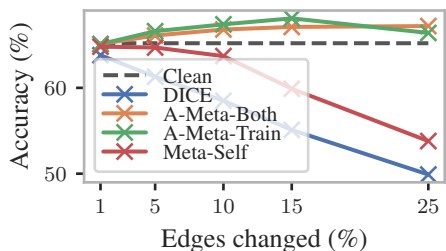

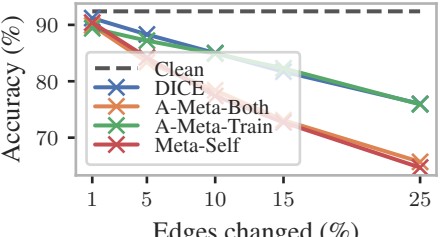

Figure 9: Change in accuracy of Deepwalk on CITESEER.

Figure 10: Change in accuracy of CLN on POLBLOGS.

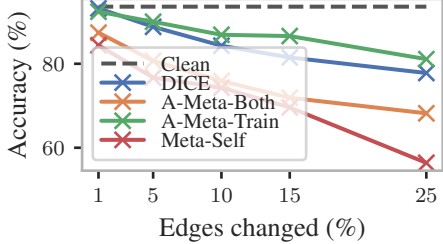

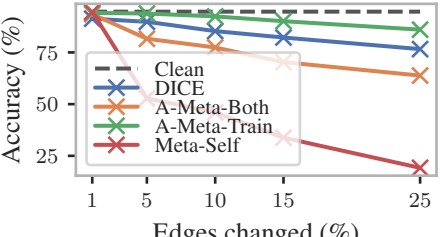

Figure 11: Change in accuracy of GCN on POLBLOGS.

Figure 12: Change in accuracy of Deepwalk on POLBLOGS.

