# OpenReview forum: "Adversarial Attacks on Graph Neural Networks via Meta Learning"
_ICLR.cc/2019/Conference_

### Official Review · AnonReviewer2 · 2018-10-27
**interesting idea and good results**

**Rating:** 6
**Confidence:** 4

**Review:**

This paper studied data poisoning attacking for graph neural networks. The authors proposed treating graph structures as hyperparameters and leveraged recent progress on meta-learning for optimizing the adversarial attacks. Different from some recent work on adversarial attacks for graph neural networks (Zuigner et al. 2018; Dai et al. 2018), which focus on attacking specific nodes, this paper focuses on attacking the  overall performance of graph neural networks. Experiments on a few data sets prove the effectiveness of the proposed approach.

Strength:
- the studied problem is very important and recently attracting increasing attention
- Experiments show that the proposed method is effective.

Weakness:
- the complexity of the proposed method seems to be very high
- the data sets used in the experiments are too small
Details:
-- the complexity of the proposed method seems to be very high. The authors should explicitly discuss the complexity of the proposed method.
-- the data sets in the experiments are too small. Some large data sets would be much more compelling.
-- Are the adversarial examples identified by the proposed method transferrable to other graph embedding algorithms (e.g., the unsupervised node embedding methods, DeepWalk, LINE, and node2vec)?
-- I like Figure 3, though some concrete examples would be more intuitive.

---

> ### Author Response · Authors · 2018-11-20
> **Re: Review 2**
>
> Dear Reviewer 2,
>
> Thank you for your constructive feedback and suggestions. We have run experiments on a larger dataset with roughly 20K nodes and found that our attacks are also successful in this scenario. You can find the results in Table 8 in Appendix F of the updated manuscript. Furthermore, we have included a discussion on the complexity of our approach in Appendix C in the updated manuscript.
>
> Regarding your question about the transferability to other graph embedding algorithms: We would like to point out that we already evaluate the impact of our attacks on DeepWalk. Our experiments show that our method’s adversarial attacks also transfer to DeepWalk.

---

### Official Review · AnonReviewer1 · 2018-11-01
**Used meta-learning by treating graph structure as hyperparameter to get the poisoned graph. Achieved reasonable results on three graph datasets.**

**Rating:** 7
**Confidence:** 4

**Review:**

This paper studies the problem of learning a better poisoned graph parameters that can maximize the loss of a graph neural network. The proposed using meta-learning to compute the second-order derivatives to get the meta-gradients seems reasonable. The authors also proposed approximate methods to compute the graph as learning parameters, which could be more efficient since the second-order derivatives are no longer computed. The experimental results on three graph datasets show that the proposed model could improve the misclassification rate of the unlabeled nodes.

The paper is well-written. It would be good if the authors could address the following suggestions or concerns:

1) The proposed attack model assumes the only the graph structure are accessiable to the attackers, which might limit the proposed model in real applications. Joint study with the graph features would be useful to convince more audience and potentially have larger impacts.

2) In the self-learning setting, in order to define l_atk, l_self is used, however, l_self is using v_u, which is the ground truth label of the test nodes based on my understanding, so this approach is using labels of the unlabeled data, which might be not applicable in real world.

3) About the action space, based on the constraints of the attacker's capability, the possible attacks will be significantly smaller than O(N^2 delta), might be O(N^delta).

4) Change 'treat the graph structure as a hyperparameter' to 'treat the graph structure tensor/matrix as a hyperparameter' would be earier to understand. And is the graph structure tensor with shape (NXN)?

5) What's the relationship between T and S? Are T in theta_T is the same as the S in G_S?

6) The title of section 4.2 is misleading. It would be better to name it as 'Greedy Computing Meta-Gradients'.

7) It lacks intuition of why define S(u,v)=delta . (-2.a_uv+1). '(-2.a_uv+1)' looks lack of intuition. Please also change 'pair (i,j), we define S(u,v)' -> 'pair (u,v)'.

8) In the experiments, what's the definition of meta-train? l_atk=-l_train?

9) In the experiments, it would be interesting to study the impact of unnoticaability constraints on the model results.

10) In figure 1, it is not surprising that when increasing the number of edges changed, the misclassification rates will increase. A graph NN considers more graph features rather than the structure is expected to show the impact of the graph structure change.

I have read the authors' detailed rebuttal. Thanks.

---

> ### Author Response · Authors · 2018-11-20
> **Re: Review 1**
>
> Dear Reviewer 1,
>
> Thank you for your detailed and constructive feedback. We have used your suggestions to improve the paper and have uploaded the updated manuscript.
>
> We would like to address each point individually here:
>
> 1) Based on your suggestion, we ran experiments on Citeseer where we use meta gradients to modify the graph structure and features simultaneously. We evaluated on GCN and CLN (DeepWalk does not use features) and we observed that the impact of the combined attacks is comparable but slightly lower (GCN: 38.6 vs 37.2, CLN: 35.3 vs 34.2; structure-only vs combined). We attribute this to the fact that we assign the same ‘cost’ to structure and feature changes, but arguably we expect a structure perturbation to have a stronger effect on performance than a feature perturbation. We have summarized these findings in Appendix E of the updated manuscript.
>
> 2) We would like to emphasize that the attack model does *not* have access to the ground-truth labels of the unlabeled nodes V_u. We use the labels of the labeled nodes to train the surrogate classification model and predict the labels \hat{C}_u of the unlabeled nodes. These labels are then treated as the ‘ground truth’ for the self-training loss L_self. Thus, the attack never uses or has access to the labels C_u of the unlabeled nodes.
>
> 3) We agree that the set of admissible attacks is significantly smaller than O(N^{2 delta}). However, since it is challenging to derive a tighter upper bound on the size of the set of admissible perturbations, we decided to use this conservative upper bound. The main point we wanted to make (which also holds for a tighter bound) is that there is an exponential growth in the number of perturbations, i.e. exhaustive search is infeasible.
>
> 4) Thank you for this suggestion. We have updated the manuscript to make this point more clear. Yes, the dimensionality of the adjacency matrix is NxN.
>
> 5) T is the number of inner optimization steps (i.e., gradient descent steps of learning the surrogate model). S is the number of meta-steps on the graph structure. We have replaced G^(S) by G^(delta) in the manuscript to avoid confusion.
>
> 6) Thank you for raising this point. We have changed the section title to ‘Greedy Poisoning Attacks via Meta Gradients’ in the updated manuscript.
>
> 7) We have changed (i,j) to (u,v). A negative gradient in an entry (u,v) means that the target quantity (e.g. error) increases when the value is decreased. Decreasing the value is only admissible for node pairs connected by an edge, i.e. we change the adjacency matrix entry from a 1 (edge) to a 0 (no edge). To account for this, we flip the sign of gradients of node pairs connected by an edge, as achieved by multiplying by (-2a_uv+1). This enables us to use the arg max operation later. Equivalently, we could compute the maximum of the gradients where there is no edge and the minimum where the nodes are connected, and then choosing the entry with the higher absolute value as the perturbation.
>
> 8) You are correct, Meta-Train uses l_atk=-l_train.
>
> 9) We have added an experiment to Appendix D showing the effect of the unnoticeability constraint (see Figure 4). As shown, even when enforcing the constraints the attacks have similar impact. Thus we conclude that the constraint should always be enforced since they improve unnoticeability while at the same time our attacks remain effective.
>
> 10) We agree that an increasing misclassification rate is expected when increasing the number of edges changed. Our intention in Figure 1 was to visualize this relationship and, more importantly, to show that our attacks consistently outperform the DICE baseline that has access to all class labels, i.e. more information than our method.

---

### Official Review · AnonReviewer3 · 2018-11-02
**Good paper of using meta-learning to solve the bilevel optimization problem in graph attacking**

**Rating:** 7
**Confidence:** 4

**Review:**

This paper proposes an algorithm to alter the structure of a graph by adding/deleting edges so as to degrade the global performance of node classification. The main idea is to use the idea of meta-gradients from meta-learning to solve the bilevel optimization problem.

The paper is clearly presented. The main contribution is to use meta-learning to solve the bilevel optimization in the discrete graph data using greedy selection approach. From the experimental results, this treatment is really effective in attacking the graph learning models (GCN, CLN, DeepWalk). However, the motivation in using meta-learning to solve the bilevel optimization is not very clear to me, e.g., what are the advantages it can offer?

Theoretically, the paper could have given some discussion on the optimality of the meta-gradient approach to bilevel optimization to strengthen the theoretical aspect. For the greedy selection approach in Eq (8), is there any sub-modularity for the score function used?

Some minor suggestions and comments:
1) please summarize the attacking procedures in the form of an algorithm
2) please have some discussion on attacking the graph attributes besides the structure
3) please have an complexity analysis and empirical evaluations of the meta-gradient computations and approximations

---

> ### Author Response · Authors · 2018-11-20
> **Re: Review 3**
>
> Dear Reviewer 3,
>
> Thank you for your constructive feedback and suggestions. We used your suggestions to improve the manuscript.
> (1+3) We have added an algorithm summary and complexity discussion to the appendix.
> (2) As Reviewer 1 also requested information about graph attribute attacks, we ran experiments on Citeseer where we use meta gradients to modify the graph structure and features simultaneously. We evaluated on GCN and CLN (DeepWalk does not use features) and we observed that the impact of the combined attacks is comparable but slightly lower (GCN: 38.6 vs 37.2, CLN: 35.3 vs 34.2; structure-only vs combined). We attribute this to the fact that we assign the same ‘cost’ to structure and feature changes, but arguably we expect a structure perturbation to have a stronger effect on performance than a feature perturbation. We have summarized these findings in Appendix E of the updated manuscript.
>
> Regarding your question about the benefit of meta-learning: Meta learning is a principle that enables us to directly tackle the bilevel optimization problem. That is, the meta gradient gives us an indication of how the value of the outer optimization problem will change when modifying the input to the inner optimization problem (i.e. the classifier training). This proves to be a very powerful principle for poisoning attacks (essentially a bilevel optimization problem) on node classification as we show in our work.

---

> > ### Comment · AnonReviewer3 · 2018-11-27
> > **the authors have addressed my concerns**
> >
> > The authors have made efforts in addressing my concerns and have improved their paper.

---

### Public Comment · (anonymous) · 2018-11-09
**Why is this problem important?**

Graph neural networks are just special cases of neural networks for classifying text (which is just a chain graph). To generate text that fools state-of-the-art classifiers one doesn't need to do much, and certainly not the method used in the paper (see e.g. the discussion in https://openreview.net/forum?id=ByghKiC5YX&noteId=B1xno5Dz6X). It is therefore quite obvious that it is even easier to fool graph neural networks, so why all the fancy methods?

---

> ### Author Response · Authors · 2018-11-09
> **Authors' response**
>
> Dear commenter,
>
> While we appreciate any constructive feedback and questions on OpenReview, we have the impression that you have not read our paper. Still, since your comment contains various incorrect claims, we address your points here:
>
> 1) Graph neural networks are NOT a special case of networks for text classification. If at all, they are generalizations. We recommend to read the broad literature on graph neural networks to clarify your confusion (references are mentioned in our paper). Here we just want to point out two important differences: (i) The neighborhood in graphs is not ordered; unlike text/images where you have before-after/left-right-up-down information. (ii) The interaction structure in graphs, i.e. the edges, is an explicit part of the data (i.e. observed) -- while in text it is NOT. Put simply: The graph structure is part of the data and, thus, can be manipulated. This is what we consider in our work.
>
> 2) You are linking to a discussion which does NOT apply to our setting. (i) It talks about text classification. (ii) The discussion you are linking to claims that text classification can easily be fooled (e.g. just simple random perturbations). Simple perturbations, however, do NOT have a strong effect on graph neural networks. This result was already clearly shown by other graph attack papers (see again the references in our paper). We also compare to strong baselines (including a random one) in our work which are consistently outperformed by our method.
>
> 3) Your statement “it is even easier to fool graph neural networks” is simply incorrect. Due to (1) you cannot make any direct conclusion from text to graphs and due to (2) it has been shown that it is NOT easy to fool graph neural networks (e.g. with random perturbations). Due to the challenging nature of achieving graph attacks, we need more advanced principles -- like the one proposed in our paper.

---

### Meta-Review · Area_Chair1 · 2018-12-19
**A novel meta-learning based approach for testing robustness of grap neural nets**

**Confidence:** 4
**Recommendation:** Accept (Poster)

**Metareview:**

 The paper proposes an method for investigating robustness of graph neural nets for node classification problem; training-time attacks for perturbing graph structure are generated using  meta-learning approach. Reviewers agree that the contribution is novel and empirical results support the validity of the approach.